# CREATOR: Tool Creation for Disentangling Abstract and Concrete Reasoning of Large Language Models

**Cheng Qian[1], Chi Han[2], Yi R. Fung[2], Yujia Qin[1], Zhiyuan Liu[1*], Heng Ji[1*]**
[1]Tsinghua University, [2]University of Illinois at Urbana-Champaign
qianc20@mails.tsinghua.edu.cn

## Abstract

Large Language Models (LLMs) have made significant progress in utilizing tools, but their ability is limited by API availability and the instability of implicit reasoning, particularly when both planning and execution are involved. To overcome these limitations, we propose CREATOR, a novel framework that enables LLMs to create their own tools using documentation and code realization. CREATOR disentangles abstract tool creation and concrete decision execution, resulting in improved performance. We evaluate CREATOR on MATH and TabMWP benchmarks, respectively consisting of challenging math competition problems and diverse tabular contents. Remarkably, CREATOR outperforms existing chain-of-thought, program-of-thought, and tool-using baselines. Additionally, we introduce the Creation Challenge dataset, featuring 2K diverse questions, to emphasize the necessity and benefits of LLMs' tool creation ability. Further research demonstrates that leveraging LLMs as tool creators facilitates knowledge transfer, and LLMs exhibit varying levels of tool creation abilities, enabling them to adapt to diverse situations. The tool creation ability revolutionizes the LLM's problem-solving paradigm, driving us closer to the next frontier of artificial intelligence. All the codes and data are released[1].

## 1 Introduction

In recent years, notable progress has been made in large language models (LLMs) like GPT-3 (Brown et al., 2020), Codex (Chen et al., 2021), PaLM (Chowdhery et al., 2022), LLaMA (Touvron et al., 2023), ChatGPT (OpenAI, 2022), and the latest GPT-4 (OpenAI, 2023). These models exhibit impressive capabilities in in-context learning, code generation, and various Natural Language Processing (NLP) tasks (Feng et al., 2020; Dong et al.,

2022). However, there are still limitations to address, such as the inability to handle up-to-date information (Yu and Ji, 2023), provide accurate mathematical results, or reason over long chains of logic (Trivedi et al., 2022; Komeili et al., 2022; Patel et al., 2021; Hendrycks et al., 2021; Lu et al., 2022b).

To overcome these concerns, researchers have explored equipping LLMs with external tools to alleviate their memory burden and enhance their expertise (Qin et al., 2023). For instance, integrating tools such as question-answering systems or web search engines enables LLMs to learn how and when to access external resources for problem-solving (Nakano et al., 2021; Schick et al., 2023). Recent studies have also incorporated additional tools for LLMs, such as GitHub resources, neural network models (e.g., Huggingface library), and code interpreters (e.g., Python interpreter), aiming to enhance their capabilities (Gupta and Kembhavi, 2022; Surís et al., 2023; Shen et al., 2023; Liang et al., 2023; Lu et al., 2023). These tools require LLMs to provide detailed plans before utilizing them to solve complex problems.

However, tool-augmented LLMs still encounter challenges (Chen et al., 2022; Gupta and Kembhavi, 2022; Schick et al., 2023; Surís et al., 2023), particularly in the following aspects. (1) **Limitation in scope**: Current approaches focus on a limited number of tools, making it difficult to find an appropriate existing tool for new problem types. (2) **Fragility in reasoning**: Given that tasks are often complex, reasoning on the fly case-by-case can be fragile to random errors, while humans can benefit from finding robust commonalities among multiple similar questions. (3) **Insufficiency in error-handling**: Current tool utilization pipelines lack automatic and specific error handling, necessitating improvements in accuracy and robustness to ensure reliable execution results.

In this paper, we propose a novel approach to ad-

---

*Corresponding author.

[1]https://github.com/qiancheng0/CREATOR

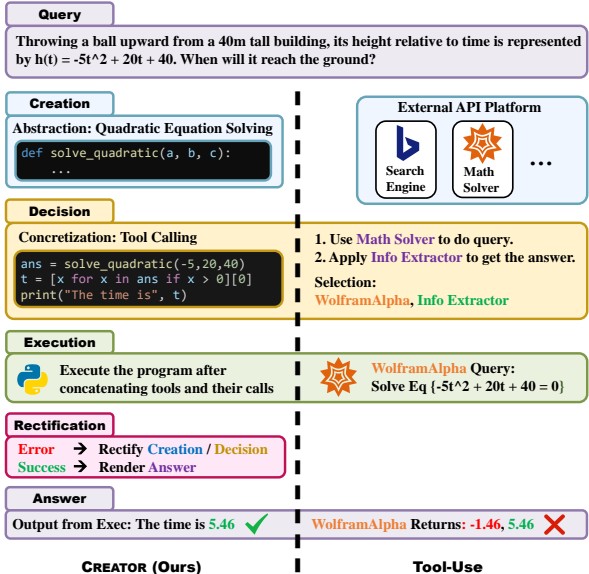

Figure 1: The difference between CREATOR and a general tool-using framework.

dress these challenges. Rather than treating LLMs as **users** of tools, we empower them to be ***creators*** of tools, enabling them to solve problems with higher accuracy and flexibility. We introduce our tool creation framework, CREATOR, which leverages LLMs' ability to create and modify tools based on the problem at hand. Figure 1 illustrates the differences between CREATOR and a general tool-using framework. While the tool-using framework focuses on reasoning to select and plan API usage, our framework emphasizes diversifying tool choices, disentangling abstract and concrete reasoning, and improving robustness and accuracy. Specifically, CREATOR consists of four stages:

- **Creation**: Create generally applicable tools with documentation and realization through abstract reasoning based on the problem.
- **Decision**: With available tools, decide when and how to use them to solve the problem.
- **Execution**: Execute the program, applying the chosen tools to solve the problem.
- **Rectification**: Make modifications to tools and decisions based on the execution result.

By introducing these four stages, we aim to better inspire the LLM's creativity and enhance the paradigm's robustness. This design sets CREATOR apart from traditional tool-using and addresses the three challenges we discussed respectively by (1) leveraging LLMs to create tools with higher generality, reusability, and variety, rather than relying on a limited number of given APIs; (2) offload-

ing the cognitive burden of LLMs and disentangling their ability to perform abstract reasoning (creation of generalizable tools) and concrete reasoning (decision-making with details); (3) utilizing code as the medium for tool creation, which is more sensitive to errors, and enabling automatic rectification of tools and decisions based on error tracebacks.

To evaluate our design's effectiveness, we test CREATOR on two existing benchmarks: MATH (Hendrycks et al.) and TabMWP (Lu et al., 2022a), as well as the Creation Challenge dataset we create. The MATH dataset contains diverse and challenging math competition problems, while TabMWP includes a wide range of tabular contexts for problem-solving. Notably, ChatGPT built on CREATOR achieves remarkable average accuracies of 59.7% and 94.7% on MATH and TabMWP respectively, surpassing the standard chain-of-thought (CoT) (Wei et al., 2022), program-of-thought (PoT) (Chen et al., 2022), and tool-using baselines by significant margins.

As existing benchmarks do not specifically evaluate tool creation, we further introduce the Creation Challenge dataset, which consists of novel and challenging problems that are inadequately solved using existing tools or code packages. This dataset highlights the necessity and advantages of LLMs' tool creation ability. In addition, we show experimental results that provide evidence of how tool creation plays a crucial role in promoting knowledge transfer across similar queries that possess common core knowledge but differ in specific scenarios. We also present case studies highlighting the varying levels of tool creation ability observed in LLMs, allowing them to better adapt to diverse problem settings.

## 2    Related Work

**Large Language Models.**    Large Language Models (LLMs) have gained attention for their impressive performance in handling various NLP tasks, following demonstrations and generating high-quality texts and codes (Brown et al., 2020; Chen et al., 2021; Chowdhery et al., 2022; Touvron et al., 2023). Prompting methods such as chain-of-thought (Wei et al., 2022), instruction-following (Wang et al., 2022b; Longpre et al., 2023; Chung et al., 2022; Touvron et al., 2023; Liu et al., 2023), and verification mechanisms (Fung et al., 2023) have been developed to guide LLMs in problem-

solving and align their behavior with human expectations. Our work builds upon these areas, incorporating them into our framework and using them as baselines for complex problem-solving.

**Tool Use and Tool Creation.** As an emerging field within NLP, the active interaction of LLMs with environments is facilitated through tools that serve as the medium (Li et al., 2023). Recent studies address constraints of LLMs, such as the limited real-time responsiveness and inaccurate calculations, by incorporating external tools (Trivedi et al., 2022; Komeili et al., 2022; Patel et al., 2021; Lu et al., 2022b). These studies augment LLMs with tools like scratch pads, search engines, QA systems, and calculators (Nye et al., 2021; Shuster et al., 2022; Schick et al., 2023) to improve task performance. More recent efforts integrate LLMs' tool-using abilities into a pipeline for task planning, tool calling, and result synthesis (Wu et al., 2023; Shen et al., 2023; Liang et al., 2023). In contrast, our work goes further by enabling LLMs to create tools instead of relying solely on existing tools. As our concurrent works, tool creation ability is also investigated under LATM framework (Cai et al., 2023) and LLM customization (Yuan et al., 2023).

**Reasoning and Execution with Program.** Reasoning with programs is an emerging field in NLP, whose goal is to leverage codes to do complicated computational reasoning instead of using natural language thoughts. Chen et al. (2022) show that code generation improves performance on math datasets, while Gao et al. (2022); Wang et al. (2022a) further demonstrate the potential of program reasoning on symbolic and algorithmic benchmarks. These efforts present a code-based chain-of-thought with linear logic but produce no enhanced tools capable of being reused or tested. As the concept of tool-using emerges, recent studies begin to incorporate code interpreters as external tools (Lu et al., 2023; Mialon et al., 2023; Wang et al., 2023). However, in CREATOR, we use code as the medium for tool creation rather than an external tool. Our framework also excels over PoT as we devise the tool creation stage, code rectification stage, and disentangle the logic in complex reasonings.

## 3 Design of CREATOR

Distinct from previous frameworks for tool-using, CREATOR leverages the tool creation ability of LLMs by incorporating four special stages: cre-

| Method | Create Tools | Utilize Tools | Apply Codes | Emphasize Reusability | Reasoning Pattern |
|---|---|---|---|---|---|
| **CoT** | - | - | - | - | *Linear* |
| **PoT** | - | - | ✓ | - | *Linear* |
| **Tool Use** | - | ✓ | ✓ | - | *Linear* |
| **CREATOR** | ✓ | ✓ | ✓ | ✓ | *Non-Linear* |

Table 1: A comprehensive comparison of CREATOR with other methods.

ation, decision, execution, and rectification, as illustrated in Figure 2. The utilization of tool creation for problem-solving is inherently straightforward and aligns with LLMs' innate ability, as illustrated later in Section 5.2. In CREATOR, the main objective of design is to instinctively better inspire their creativity, and facilitate more effective use of it.

Previous CoT and PoT methods mainly apply linear reasoning to solve target problems, and their task-solving process lacks reusability. However, the tools created in CREATOR can be transferred to solve other queries, and the rectification stage incorporated makes the reasoning process non-linear. We present a comprehensive comparison between CREATOR and other methods in Table 1.

### 3.1 Creation Stage

**Implementation Details.** In the creation stage of CREATOR, we explicitly instruct LLMs with demonstrative examples to create tools and documentation to solve the problem. The general prompt text form is "###Instruction [INSTRUCTION]\n [EXAMPLE 1]\n [EXAMPLE 2] ...". Here the instruction text "[INSTRUCTION]" describes the goal and format of the output. Each demonstration "[EXAMPLE x]" follows format "### Question [QST]\n ### Tool [TOOL]". Each [TOOL] contains documentation text as code comments. A detailed example of prompt text is shown in Figure 14.

To get these demonstrations, we make a fixed set of demonstrations in advance and use them subsequently for each task. In specific, we randomly select a subset from the training set and prompt the LLM with the text instruction for tool creation for each query. We then correct the errors in these generations (if any) and remove verbose explanations before using them. Although the demonstrations are from the same task as the test queries, they are not required to be semantically similar to test queries, as the main purpose is only to inspire the

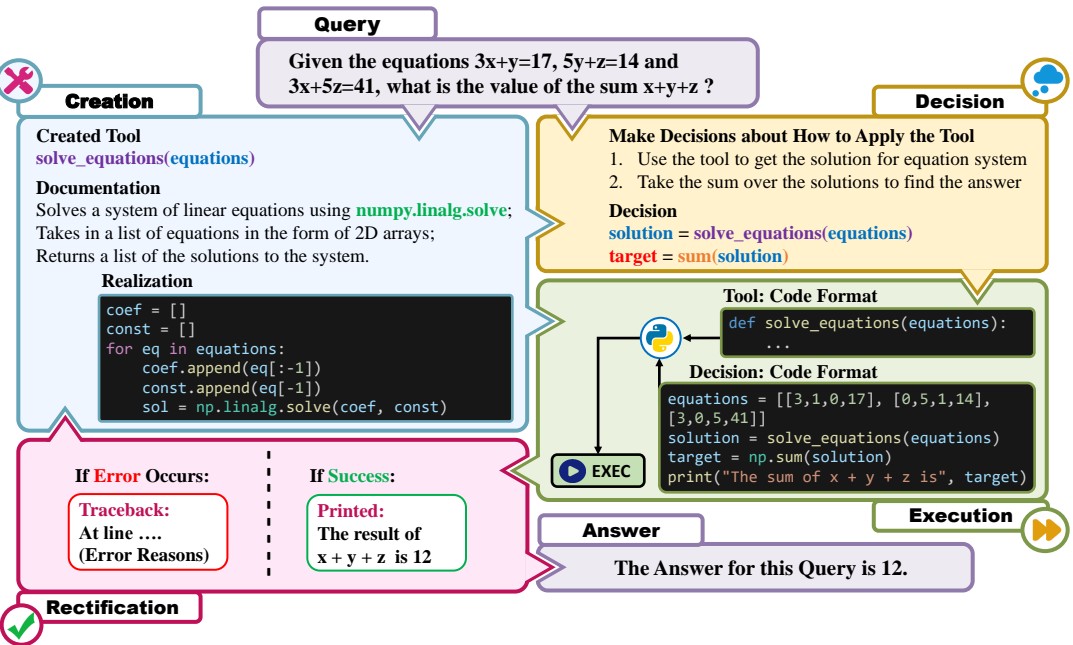

Figure 2: Overview of our CREATOR framework with four stages: Creation, Decision, Execution, and Rectification. With an LLM like ChatGPT, we successfully leverage its tool creation ability with code as the medium. In each stage we apply instructions and demonstrations in prompts, shown in Figures 14 to 16 in Appendices.

LLM's creativity and regulate its output format.

**Ability of Abstract Reasoning.** The core importance of the tool creation stage is to trigger LLM's ability to employ abstract thinking to alleviate the burden of reasoning during later stages. When LLMs create tools, they effectively use abstraction to address a particular problem type, necessitating a focus on the inherent characteristics of the problem rather than the specific numerical details. For example, in Figure 2, the LLM concentrates solely on recognizing the intrinsic nature of the problem and creates a tool for solving a three-variable equation system, disregarding all the numerical details and the specific expression being queried.

### 3.2 Decision Stage

**Implementation Details.** Similar to the creation stage, we instruct LLMs with demonstrations to decide how to use tools with the same prompt text form. Each demonstration "[EXAMPLE x]" follows "### Question [QST]\n ### Tool [TOOL]\n ### Solution [SOL]", where [SOL] represents the LLM's decision tool calls in code format. We also derive a fixed demonstration set the same way as in the creation stage, only that the LLM is now prompted to call the given tools instead of creating them, and to print out the final answer with any important information through "*print(…)*" in

codes. This [INSTRUCTION] applies both to get demonstrations and to conduct test-time inference, which ensures that the LLM's answer can be easily extracted from printed outputs in subsequent stages. A detailed prompt text example is shown in Figure 15.

**Ability of Concrete Reasoning.** The decision stage necessitates the LLM's meticulous attention to rules and details for problem-solving, which we refer to as concrete reasoning. In Figure 2, the solution obtained from the tool needs to be summed for the final answer. This requires the LLM to understand the tool's outputs and relate them to the specific query to make an informed decision and derive the correct answer finally. By separating creation from the decision, CREATOR disentangles two phases of the LLM's abilities, which facilitates a smoother elicitation of different aspects of knowledge and improves task performance.

### 3.3 Execution Stage

The execution stage takes the information from previous stages to execute the tool leveraging the code interpreter. We do not apply the LLM in this stage, and the created tools and the LLM's decision are concatenated into a cohesive code block for execution. The tool is encapsulated within a function in the code block, and the LLM's decision calls it

for problem-solving. During execution, we capture any outputs printed (as we have instructed the LLM in the decision stage) or errors encountered (by intercepting error messages in a sub-process). These information serve as inputs for subsequent stages to determine whether an answer can be obtained or rectifications are needed.

### 3.4 Rectification Stage

**Implementation Details.** During the rectification stage, CREATOR has two different options based on the information passed into it. If an error occurs, then the LLM is prompted with demonstrations to rectify the error. Applying a similar prompt format as before, the format of demonstrations "[EXAMPLE x]" now changes to "### Question [QST]\n ### Original [ORI]\n ### Error [ERR]\n ### Rectification [REC]", where we provide the original tool implementation and calling decision in [ORI], offer the error tracebacks [ERR], and concatenate natural language reasoning on the error with the rectified code in [REC]. A detailed illustration of the prompt text is shown in Figure 16.

If the execution is successful, then the answer will be extracted from the captured model's output and compared to the standard answer to measure accuracy.

**Significance.** During the rectification process, we provide the LLM with error tracebacks, which offer crucial information for it to identify the error's location and causes. Armed with this guidance, the LLM can recover from previous mistakes, adjust its reasoning process, and attempt to solve the problem once again. Subsequent experiments will demonstrate how the inclusion of rectification significantly improves the performance of CREATOR. The success of the rectification stage also showcases the LLM's ability to recognize misconceptions and self-correct.

## 4 Experiments

To evaluate the effectiveness of CREATOR, we conduct experiments on two established benchmarks: MATH (Hendrycks et al.) and TabMWP (Lu et al., 2022a). Additionally, we perform experiments on a newly introduced dataset, Creation Challenge, comprising 2K diverse questions that are inadequate to solve using existing tools or code packages. This enables us to further demonstrate the necessity and advantages of the LLM's tool creation ability.

### 4.1 Experimental Setup

**Settings.** We select ChatGPT as the base model for all methods due to its exceptional capabilities in code generation, decision-making, and logical reasoning. Refer to Appendices A.1 for more details. We evaluate CREATOR on two existing datasets: TabMWP, which includes diverse table-related problems, and MATH, consisting of challenging math competition problems. We apply them as they are representative in terms of diversity in data format and difficulty. We also assess the performance of our framework on Creation Challenge, comprising 2K data points, to explore the impact of tool creation hints on the LLM's performance. Refer to Appendices A.2 for more details.

**Baselines.** We compare CREATOR against four types of baselines to demonstrate its effectiveness:

- **Vanilla LLM w/ and w/o CoT**: The Vanilla LLM with CoT employs linear reasoning to solve problems, while Vanilla LLM without CoT directly generates the answer.

- **PoT**: The LLM utilizes a program to reason through the problem step by step. Besides, we also incorporate rectification into PoT as a stronger baseline for a fair comparison.

- **Tool Use**: The LLM utilizes the WolframAlpha API as a general-purpose tool specialized in calculations. It's a fair external tool as all queries require numerical reasoning to some extent.

- **CREATOR-Entangled**: The LLM combines the creation and the decision stage in CREATOR instead of disentangling them, which serves as a special baseline for ablation study.

### 4.2 Creation Challenge

Existing benchmarks are not originally designed to evaluate tool creation, thus unable to fully showcase the necessity and advantages brought by the LLM's tool creation ability. Therefore, we introduce Creation Challenge to test the LLM's problem-solving skills under new scenarios, without existing tools or code packages that can be directly applied. Refer to Appendices B.1 for details about the data format and construction process.

**Evaluation** The components of the standard created tool in each data point of Creation Challenge can serve as valuable hints for the LLM's tool creation. Therefore, we extend our experiments on Creation Challenge to assess the LLM's tool

| Method | Setting | Algebra | Counting & Probability | Geometry | Itmd. Algebra | Number Theory | Pre-Algebra | Pre-Calculus | Average (weighted) |
|---|---|---|---|---|---|---|---|---|---|
| **Vanilla** | *w/o CoT* | 25.7 | 25.8 | 22.4 | 13.9 | 18.5 | 40.9 | 21.8 | 25.3 |
| | *w/ CoT* | 50.9 | 36.1 | 24.5 | 17.5 | 23.2 | 58.6 | 16.7 | 37.9 |
| **PoT *(w/o Rec.)*** | *w/o CoT* | 58.2 | 48.5 | 35.4 | 25.8 | 53.1 | 66.8 | 25.0 | 49.8 |
| | *w/ CoT* | 54.0 | 47.8 | 32.5 | 22.3 | 48.9 | 64.5 | 19.9 | 46.5 |
| **PoT *(w/ Rec.)*** | *w/o CoT* | 63.8 | 51.9 | 35.9 | 28.6 | 59.2 | 70.0 | 28.2 | 53.9 |
| | *w/ CoT* | 61.4 | 48.8 | 34.6 | 23.7 | 54.5 | 67.6 | 34.6 | 51.2 |
| **Tool Use** | *w/o CoT* | 47.3 | 35.1 | 27.0 | 20.5 | 30.8 | 56.8 | 31.4 | 39.0 |
| | *w/ CoT* | 55.3 | 37.8 | 28.7 | 20.5 | 34.8 | 61.8 | 26.9 | 43.0 |
| **CREATOR -Entangled** | *w/o Demo.* | 58.0 | 53.3 | 34.2 | 21.8 | 55.7 | 63.4 | 33.3 | 49.6 |
| | *w/o CoT* | 64.1 | 55.7 | 35.9 | **42.7** | 61.6 | 69.0 | **37.2** | 57.2 |
| | *w/ CoT* | 62.7 | 50.9 | 33.8 | 31.4 | 61.4 | 68.7 | 31.4 | 54.0 |
| **CREATOR (ours)** | *w/o Demo.* | 66.6 | 53.6 | 33.8 | 29.4 | 59.8 | 68.7 | 34.6 | 54.9 |
| | *w/o CoT* | **71.5** | 55.3 | **41.4** | 41.9 | 60.4 | **71.7** | 35.3 | **59.7** |
| | *w/ CoT* | 63.1 | **58.1** | 34.6 | 35.0 | **61.8** | 69.7 | 32.1 | 55.7 |

Table 2: The accuracy (%) on the test set of MATH dataset leveraging ChatGPT. *Rec.* represents Rectification.

creation ability with varying levels of hint utilization. We encourage future research to explore the dataset's potential through more flexible usage.

### 4.3 Experimental Results

We present the results on MATH, TabMWP, and Creation Challenge respectively in Tables 2 to 4. CREATOR achieves an accuracy of 59.7%, 94.7%, and 75.5% respectively on three tasks, surpassing all the best performance in baselines by large margins. To illustrate CREATOR's advantage, we present a case study showing how it's better than Tool Use in Figure 3A. For all tasks, disentangling the creation and decision stages generally results in better performance, compared to CREATOR-Entangled. For Creation Challenge, we also observe that hints of tool creation can raise the performance up to 18.7%. We will further analyze the reasons for this improvement in Section 4.4.

### 4.4 Results Analysis

**CoT Incompatible with Codes.** Table 2 shows the LLM's performance on MATH problems decreases consistently when applying CoT under PoT method and CREATOR, and the opposite trend is observed for TabMWP. We attribute this difference to the inherent incompatibility between natural language reasoning and program-based reasoning on challenging problems. MATH problems involve intricate calculations and diverse reasoning paths, leading to conflicts between natural language and programming approaches. When CoT is used, the LLM tends to generate programs following natural language reasoning, which compromises the coherence and unique advantages of programming. In

| Method | Setting | Accuracy | Successful Execution |
|---|---|---|---|
| **Standard** | *w/o CoT* | 68.2 | 99.1 |
| | *w/ CoT* | 75.2 | 99.3 |
| **PoT *(w/o Rec.)*** | *w/o CoT* | 80.6 | 98.5 |
| | *w/ CoT* | 80.0 | 91.2 |
| **PoT *(w/ Rec.)*** | *w/o CoT* | 81.2 | 99.7 |
| | *w/ CoT* | 87.3 | 100 |
| **Tool Use** | *w/o CoT* | 77.6 | 100 |
| | *w/ CoT* | 79.6 | 100 |
| **CREATOR-Entangled** | *w/o CoT* | 91.6 | 100 |
| | *w/ CoT* | 93.5 | 99.9 |
| **CREATOR (ours)** | *w/o CoT* | 90.5 | 99.7 |
| | *w/ CoT* | **94.7** | 100 |

Table 3: The accuracy (%) on the test set of TabMWP dataset leveraging ChatGPT. *Successful Execution* indicates whether the LLM provides a valid final answer through words or codes within the rectification limit.

Figure 3B. we show the adoption of brute-force algorithms and straightforward calculations when CoT is not applied yields higher accuracy.

In contrast, TabMWP involves simpler calculations and more straightforward reasoning paths, promoting consistency between natural language and programming reasoning. Therefore, the application of CoT enhances performance in these cases. We present more case studies to illustrate it in Appendices C.1.

**CREATOR is Robust to Challenges.** Figure 4 illustrates the performance of the LLM in relation to difficulty. CREATOR outperforms all the baselines for both tasks and achieves higher accuracy, particularly for difficult problems. This provides compelling evidence that CREATOR exhibits greater resilience to challenges.

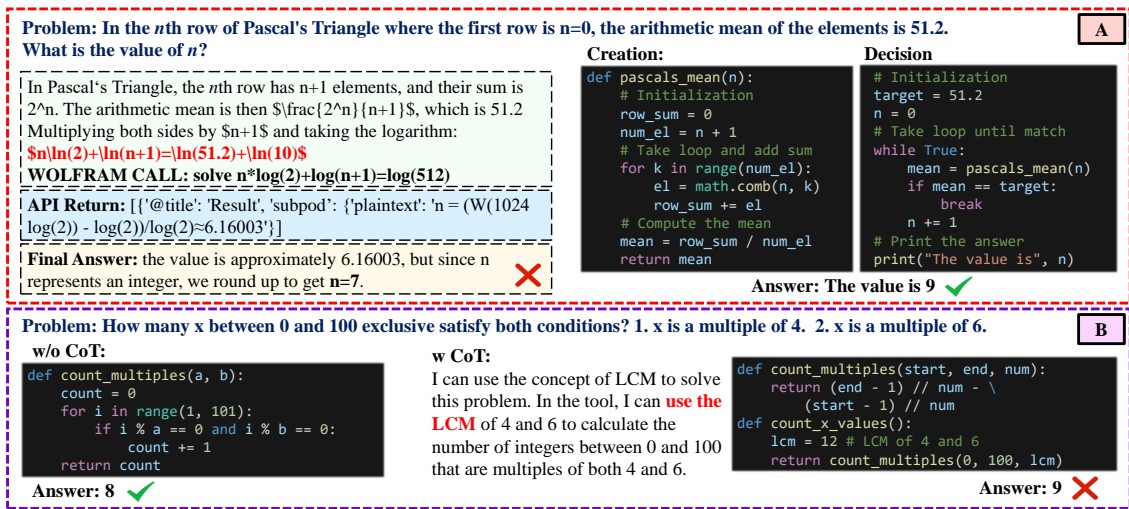

**Problem: In the *n*th row of Pascal's Triangle where the first row is n=0, the arithmetic mean of the elements is 51.2. What is the value of *n*?**

In Pascal's Triangle, the *n*th row has n+1 elements, and their sum is $2^n$. The arithmetic mean is then $\frac{2^n}{n+1}$, which is 51.2
Multiplying both sides by $n+1$ and taking the logarithm:
$n\ln(2)+\ln(n+1)=\ln(51.2)+\ln(10)$
**WOLFRAM CALL: solve n*log(2)+log(n+1)=log(512)**

**API Return:** [{'@title': 'Result', 'subpod': {'plaintext': 'n = (W(1024 log(2)) - log(2))/log(2)≈6.16003'}]

**Final Answer:** the value is approximately 6.16003, but since n represents an integer, we round up to get **n=7.** ❌

Creation:
```
def pascals_mean(n):
    # Initialization
    row_sum = 0
    num_el = n + 1
    # Take loop and add sum
    for k in range(num_el):
        el = math.comb(n, k)
        row_sum += el
    # Compute the mean
    mean = row_sum / num_el
    return mean
```

Decision
```
    # Initialization
    target = 51.2
    n = 0
    # Take loop until match
    while True:
        mean = pascals_mean(n)
        if mean == target:
            break
        n += 1
    # Print the answer
    print("The value is", n)
```
**Answer: The value is 9** ✓

**Problem: How many x between 0 and 100 exclusive satisfy both conditions? 1. x is a multiple of 4.  2. x is a multiple of 6.**

w/o CoT:
```
def count_multiples(a, b):
    count = 0
    for i in range(1, 101):
        if i % a == 0 and i % b == 0:
            count += 1
    return count
```
**Answer: 8** ✓

w CoT:
I can use the concept of LCM to solve this problem. In the tool, I can **use the LCM** of 4 and 6 to calculate the number of integers between 0 and 100 that are multiples of both 4 and 6.
```
def count_multiples(start, end, num):
    return (end - 1) // num - \
        (start - 1) // num
def count_x_values():
    lcm = 12 # LCM of 4 and 6
    return count_multiples(0, 100, lcm)
```
**Answer: 9** ❌

Figure 3: In subfigure A, we show an example in which Tool Use reasoning (left) fails, while CREATOR (right) solves successfully as it derives a new tool for the novel question. In subfigure B, we present a case comparing the answer given by CREATOR with and without CoT. Challenging problems in MATH cause conflicts between language and program reasoning.

| Method | Setting | Accuracy | Successful Execution |
|---|---|---|---|
| **Standard** | *w/o CoT* | 27.9 | 94.9 |
| | *w/ CoT* | 32.7 | 99.1 |
| **PoT (*w/o Rec.*)** | *w/o CoT* | 59.2 | 93.5 |
| | *w/ CoT* | 60.7 | 95.7 |
| **PoT (*w/ Rec.*)** | *w/o CoT* | 61.1 | 98.3 |
| | *w/ CoT* | 62.0 | 98.9 |
| **CREATOR-Entangled (*w/o CoT*)** | *no hint* | 64.5 | 99.2 |
| | *utility hint* | 65.8 | 99.3 |
| | *all hint* | 75.3 | 99.5 |
| **CREATOR (ours) (*w/o CoT*)** | *no hint* | 63.8 | 98.7 |
| | *utility hint* | 67.2 | 99.1 |
| | *all hint* | **75.7** | 99.5 |

Table 4: The accuracy (%) on the Creation Challenge test set leveraging ChatGPT. *No hint* represents normal CREATOR framework. *Utility hint* provides hints about the utility of the tool, while *all hint* offers additional hints about the possible inputs and outputs of the tool.

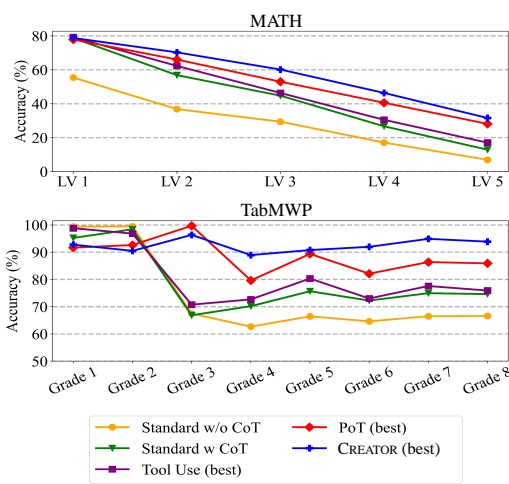

Figure 4: Comparison of the accuracy of baselines and CREATOR *w.r.t.* problem difficulty.

**Rectification Raises Performance.** Figure 5 demonstrates the improvement in the LLM's performance achieved through the application of the rectification stage. Results show rectification can increase the accuracy by approximately 10% of the original value, which proves the necessity and rationality of establishing this stage.

**Influential Factors of Tool Creation.** Tables 2 to 4 highlight two crucial factors affecting the LLM's performance. (1) **Separation of Creation and Decision**: The separation of these two stages inherently represents the disentanglement of the LLM's abstract and concrete reasoning, which

leads to improved performance. (2) **Availability of Hints**: In practical scenarios, guidance is often necessary to harness the LLM's behavior when creating tools. We demonstrate that providing more detailed hints can significantly improve the LLM's performance, as they enable easier implementation of desired tools and eliminate uncertainty and misdirections in CoT or tool documentation.

## 5 Further Analysis

In this section, we further show the advantages brought by the LLM's tool creation ability and use case studies to demonstrate different aspects of this ability, which enables them to tackle challenges

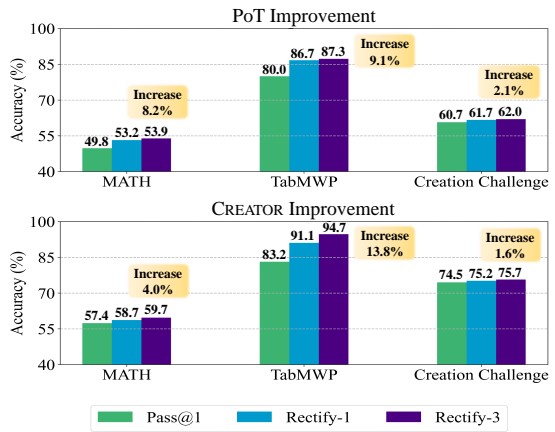

Figure 5: The improvement brought by rectification on both PoT and CREATOR. *Rectify-N* denotes enabling N rounds of rectifications.

| | |
|---|---|
| Set of Queries, Count | 100 |
| Data Pieces, Count | 300 |
| Tool Create Normal, Acc. | 63.0% |
| Tool Create with Transfer, Acc. | 78.3% |
| Increase of Acc. | **15.3%** |
| Sets Worse with Transfer | 2 / 100 |
| Sets Better with Transfer | **39 / 100** |

Table 5: Results of tool transfer experiment. Tool transfer improves accuracy by up to 15.3%.

with more flexibility and less reasoning burden.

## 5.1 Facilitation of Knowledge Transfer

One of the main purposes of tool creation lies in its reusability. The content of tools represents the abstraction of knowledge concepts, so the creation of one tool may help solve problems of various scenarios that share the same core concept. For instance, a keyword-extraction tool created for sentiment analysis can be applied to other tasks like document categorization and topic modeling, as they all require the identification and extraction of relevant keywords for problem-solving. By utilizing the knowledge and logic embedded in the tool, the LLM can transfer its understanding to solve similar problems efficiently with higher performance.

**Settings.** To validate our hypothesis, we construct a small set of questions with 300 data points, detailed in Appendices B.2. We divide data points into 100 sets, where all three queries in one set share the same core knowledge concept (key methodology that is universally applicable) but differ in scenario (problem background and specific details inquired).

Similar to previous experiments, we use Chat-GPT as the base LLM with unchanged detailed settings. We first test all the problems under the normal CREATOR framework respectively. Then, we test if the correct tool created under one scenario could be applied to the other two, and again test the LLM's performance.

**Results Analysis.** The statistics are presented in Table 5. Through the application of transferred tools, the LLM's accuracy can be raised by 15.3%. Further analysis shows that 39 sets of queries are positively influenced by this transfer, which highlights **the tool creation ability of the LLM can facilitate knowledge transfer, leading to better performance on clusters of problems that share similar core concepts.**

## 5.2 Different Levels of LLM's Tool Creation

We discover in experiments that LLM can create tools in different levels without special guidance, which affirms creativity is LLM's intrinsic emerging ability. By inspecting the created tools, we find that they can be categorized into three levels, which provides guidance and reference for future development.

**1. Enhancement of Existing Tool.** First, LLMs demonstrate the capability to enhance existing tools by encapsulating an existing tool or API and repurposing it to serve different needs. The first case of Figure 9 shows how LLM wraps an existing weather query API into a new tool that calculates the average temperature.

**2. Concatenation of Multiple Tools.** Second, the LLM can create new tools by organizing multiple APIs into a pipeline, enabling it to fulfill specific purposes. The second case in Figure 9 shows how the LLM calls two existing APIs three times in the new tool for problem-solving.

**3. Hierarchical Tool.** Third, the LLM can create tools with a clear hierarchy, which establishes clear caller-callee relationships among tools and reduces the burden of repetitive reasoning. The third case in Figure 9 illustrates a hierarchical structure where the first tool serves as the callee, while the second tool primarily solves the problem.

## 6 Conclusion

We propose the concept of automatic tool creation through LLMs and empirically devise CREATOR

that harnesses the capabilities of LLMs as tool creators. By disentangling LLM's abstract and concrete reasoning, CREATOR enables clearer logic and enhances overall performance. Through comprehensive evaluations on established benchmarks and Creation Challenge, we demonstrate the superiority and indispensability of CREATOR compared to existing CoT, PoT, and tool-using approaches. We anticipate our study will inspire the development of more sophisticated AI systems leveraging LLM's tool creation potential.

## Limitations

Our experiment is limited to two established benchmarks, MATH and TabMWP, along with our newly introduced dataset, Creation Challenge. However, it is crucial for future research to expand the application of our framework to encompass a broader array of tasks. This will enable a comprehensive assessment of the generalizability of our results, going beyond the scope of our current investigation.

Furthermore, our demonstration of the LLM's potential in tool creation is limited in scope. For instance, the current LLM is also capable of creating tools even to build a full project pipeline, but the execution ability and correctness of its creation still lack proper evaluations and remain questionable. It is incumbent upon future research to delve deeper into the boundaries of LLM's capabilities and establish clear limits regarding its tool creation potential.

## Ethics Statement

We consider the following research issues in this paper:

- **Privacy** involves safeguarding sensitive information and preventing its unauthorized disclosure. With respect to our framework, privacy becomes a concern when certain stages require demonstration examples and clear instructions, which may inadvertently contain sensitive information, or intentionally designed to prompt the LLM to leak privacy. Thus, it is crucial to ensure that personal or sensitive information is not disclosed to the closed-source LLM, and the private information or knowledge about tool creation in the closed-source LLM should be well-protected.

- **Fairness** in AI aims to ensure the outputs and decisions made by AI systems do not perpetuate existing biases or discriminations. When

creating tools, care must be taken to mitigate biases in the demonstrations and instructions, monitor the tool's performance across stages, and address any disparities that may arise in the whole generation or rectification process.

- **Transparency** involves making AI systems and their processes understandable and interpretable. When the language model creates tools under our framework, it's essential to have transparency regarding how those tools are developed. Developers should document any biases or limitations associated with the tools created, understand the strengths and weaknesses of the tools and how the decision is reached, and make informed decisions about their application.

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

# Appendices

## A  Details about Experiment

### A.1  Model Details.

We employ GPT-turbo-3.5 as the base model for all our experiments. The maximum generation length for all experiments is set to 512, and a temperature of 0.3 is chosen to encourage deterministic generations while maintaining a certain degree of diversity, particularly during the creation of tools.

### A.2  Dataset Details.

For both the MATH and TabMWP datasets, we evaluate questions that have numerical value answers (e.g. integers or decimals). This is due to the complicated format matching problems (e.g. matching of matrices as answers) that may cause bias. The tested questions cover approximately 80% of all and maintain high diversity, making our results still representative. We are planning to update our results on all MATH questions applying post-processing soon [2], but some matching problems are still hard to solve.

The MATH dataset consists of seven math competition problem domains, namely algebra, counting and probability, geometry, intermediate algebra, number theory, pre-algebra, and pre-calculus. Each domain is evaluated separately, and the final metric is computed as the weighted average score. The TabMWP dataset includes a wide range of table information and problems of different difficulty levels, spanning from grade one to grade eight.

## B  Details about New Datasets

### B.1  Creation Challenge Details

We begin by constructing a seed dataset that involves novel settings and unconventional reasoning processes. Subsequently, we utilize the Text-Davinci-003 model to expand the dataset in an iterative manner. By random sampling from the seed data, we encourage the variety and novelty in the problems and their reasonings.

Figure 6 illustrates a sample query and its corresponding solution. Each data entry comprises the problem statement, a standard created tool that can be utilized (including utility, input, output, and realization), a tool-calling decision, and a final answer.

[2] https://github.com/openai/prm800k/blob/main/prm800k/grading/grader.py

Figure 6: An example query and its solution provided in the Creation Challenge dataset.

### B.2  Tool Transfer Dataset Details

We create 300 data points in total and divide them into sets of three. Each set of three queries contains three corresponding answers, one standard tool that could be applied in all three scenarios to solve the problem, and three decisions about how to use the tool respectively. Similar to the construction of Creation Challenge, we manually write the seed data, which includes five sets of queries used as examples to show the format of each data point, sample demonstration examples from these seeds, and leverage the Text-Davinci-003 to create more data iteratively.

We present a sample set from the tool transfer dataset we curate in Figure 7. In the set, three different scenarios are provided, with each one consisting of a query, a sample decision, and an answer (not listed). Though the scenarios seem unrelated, they share the same core knowledge which can be transferred. In this case, the core knowledge is the calculation of profit. We also provide an example tool that can be applied to all these three scenarios with a corresponding introduction. Note that each set we define actually represents three data points.

## C  More about Experimental Findings

### C.1  CoT Incompatible with Code

In this section, we will provide more cases to further illustrate our arguments made in Section 4.4 about the conflicts between natural language thoughts and program thoughts. We con-

**Sample Tool (Common For 3 Scenarios)**

The tool is used to calculate the optimal number of units to produce to maximize profit for a manufacturing company. It takes into account the fixed costs, variable costs, selling price, and demand for the product. The function uses the formula *Profit = (Selling Price \* Quantity) - (Variable Cost \* Quantity) - Fixed Cost* to calculate the profit and returns the optimal quantity to produce.

```python
def calculate_optimal_units(selling_price, variable_cost,
    fixed_cost, demand):
    # Calculate the profit for each quantity
    profits = []
    for quantity in range(1, demand+1):
        profit = (selling_price * quantity) - (variable_cost *
                  quantity) - fixed_cost
        profits.append(profit)
    # Find the quantity that maximizes profit
    optimal_quantity = profits.index(max(profits)) + 1
    # Return the optimal quantity
    return optimal_quantity
```

**Scenario 1: Pricing Strategy**

A company produces a product that has a fixed cost of $20,000, a variable cost of $10 per unit, and a demand of 10,000 units. The company wants to maximize profit and is considering two pricing strategies. The first strategy is to sell the product at $30 per unit, and the second strategy is to sell the product at $35 per unit. What is the optimal pricing strategy for the company?

```python
fixed_cost = 20000
variable_cost = 10
demand = 10000
# Strategy 1: Selling the product at $30 per unit
selling_price_1 = 30
optimal_quantity_1 = calculate_optimal_units(selling_price_1,
variable_cost, fixed_cost, demand)
profit_1 = (selling_price_1 * optimal_quantity_1) -
(variable_cost * optimal_quantity_1) - fixed_cost
# Strategy 2: Selling the product at $35 per unit
selling_price_2 = 35
...
# Determine the optimal pricing strategy
if profit_1 > profit_2:
    ...
```

**Scenario 2: Production Planning**

A manufacturing company produces a product that has a fixed cost of $10,000, a variable cost of $5 per unit, and a selling price of $20 per unit. The company can sell up to 5,000 units of the product at this price. What is the optimal number of units to produce to maximize profit?

Figure 7: An example data point in tool transfer dataset. We provide three scenarios sharing the core knowledge and a sample tool that all three scenarios can utilize.

trast two additional cases sourced respectively from MATH and TabMWP in Figure 8.

In the case of MATH, the ambiguity of "string manipulation" mentioned in natural language thoughts leads the model to create the tool that finds the hundredth digit in a hard-coding manner, while pure code generation in creating tools can avoid this problem.

Conversely, for TabMWP, CoT helps tool creation by avoiding unnecessary complexities in sim-

ple problem-solving. In the second case, the natural language thoughts indicate clearly that only simple multiplication should be done, while pure code generation is trapped in a complex and chaotic logic that is prone to error.

These two cases further validate the conflicts between natural language thoughts and program thoughts, especially for challenging problems which may possess multiple reasoning paths that differ in suitability for code and natural language.

## C.2 Different Levels of Tool Creation

We present in this section more details about the different levels of tool creation mentioned in Section 5.2. We present three cases in Figure 9.

The enhancement of existing tools in tool creation is presented in the first case. After the query, the LLM is given an existing API that could be called for a fixed purpose. This mimics the scenario in the real world where an API document is given to let one fulfill a particular purpose. At this level, the LLM learns how to create tools first by comprehensively understanding the existing tool and then transferring this knowledge to a new problem scenario. In this case, the LLM learns how temperature should be averaged across several days, and subsequently creates a tool that solves the problem.

The concatenation of multiple tools in tool creation is presented in the second case. In this case, the LLM is given several tools to solve the problem, but the usage of each tool is rather simple to follow. This level of tool creation requires the LLM to plan the use of tools in a logical way and organize them with clear logic. Instead of how to call tools to serve a different purpose, this level also illustrates the LLM's excellent ability in implementing a pipeline to solve specific queries through tool creation.

The hierarchy of tool creation is presented in the third case. This not only is the most common phenomenon that we observe in the experiment but also represents the most advanced aspect of the LLM's reasoning potential. By creating tools with a clear hierarchy, the LLM is successful in offloading more reasoning burdens and thus solving the problem with higher accuracy. In this case, *is_prime* represents only a "sub-tool", while the main tool solves the problem with more ease by calling it to help count the valid numbers.

Overall, the presented case studies provide valuable insights into the tool creation abilities of

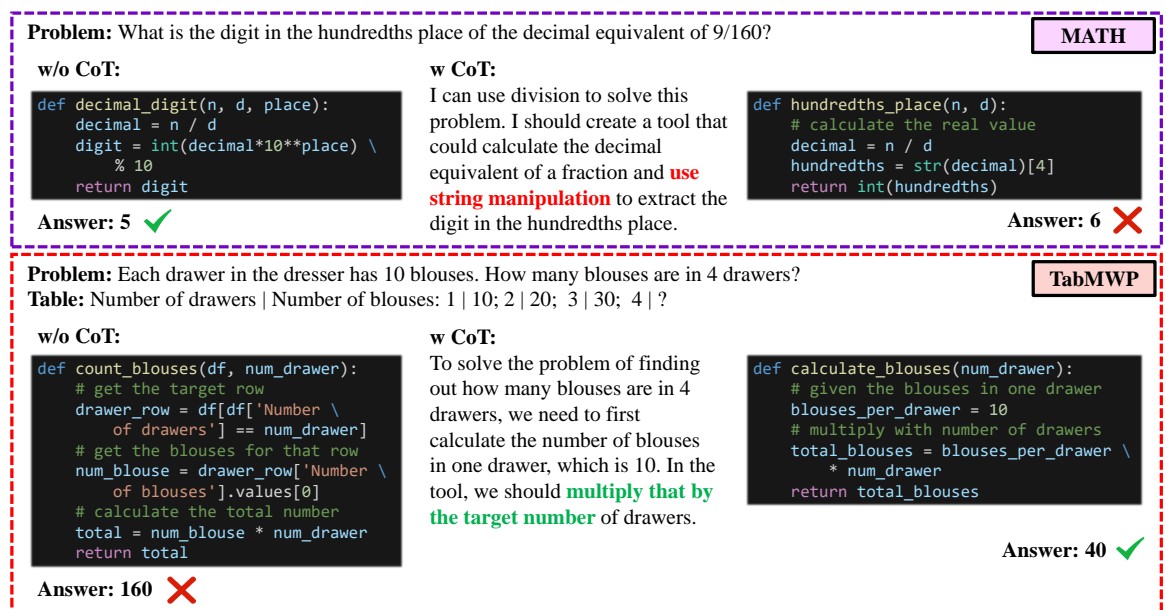

Figure 8: We present two more cases to illustrate the conflicts between program thoughts and natural language thoughts.

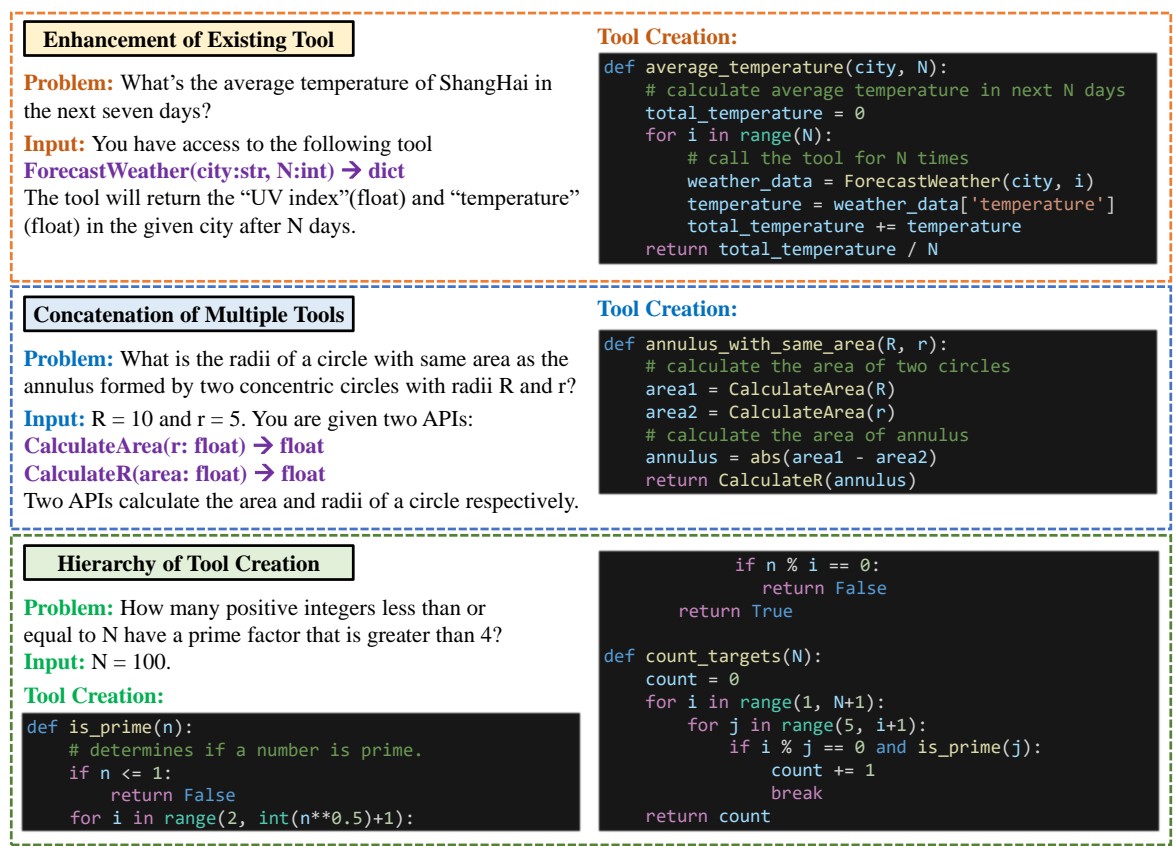

Figure 9: We present three cases to illustrate the idea of the LLM's tool creation from different levels.

LLMs. However, it is important to acknowledge that these studies offer only a glimpse into the vast potential of LLMs in this domain. We encourage future research to explore and harness the full extent of LLMs' tool creation capabilities, further pushing the boundaries of what can be achieved.

## D  Prompting Details

All the methods we present in our main experiments need prompting to formalize the LLM's response and better inspire its ability.

**Prompting of CREATOR.**  We present in Figures 14 to 16 the general prompting format and the formats of demonstrative examples, as detailed in details in 3. For the creation stage, decision stage, and rectification stage, we apply demonstration examples to enhance the LLM's abstract and concrete reasoning ability, while the execution stage intrinsically is unrelated to the LLM. We present one demonstrative example about a query in MATH, but other tasks including TabMWP and Creation Challenge also follow this prompting format.

**Prompting of Baselines.**  Besides CREATOR, we also apply demonstrative examples in prompting the ChatGPT's CoT, PoT, Tool Use abilities respectively, presented in Figures 10 to 13. Similar to the prompting of CREATOR, these prompt formats apply to all tasks in the main experiments, including evaluation on MATH, TabMWP, and Creation Challenge.

Specifically, We separate Tool Use into two parts, the first one aiming to inspire the LLM's ability to call WolframAlpha properly, and the second one aiming to prompt the LLM to retrieve the final answer. For CREATOR setting, the prompts are separated according to different stages. Note that the execution stage does not need prompting.

---

**### Instruction**
You are given a math question.
You should think step by step to solve the problem.
Please put your final numerical answer after "Final Answer:"

**### Question**
Point $P$ lies on the line $x= -3$ and is 10 units from the point $(5,2)$. Find the product of all possible $y$-coordinates that satisfy the given conditions.
**### Thoughts and Answer**
Let the y-coordinate of point $P$ be denoted by $y$. Since $P$ lies on the line $x=-3$, we know that the coordinates of $P$ are $(-3,y)$.
Since $P$ is 10 units from the point $(5,2)$, we can use the distance formula to write an equation:
$\sqrt{(-3-5)^2+(y-2)^2}=10$
Simplifying the equation gives us:
$(y-2)^2+64=100$
$(y-2)^2=36$
$y-2=\pm6$
$y=8$ or $y=-4$
Therefore, the possible $y$-coordinates that satisfy the given conditions are $8$ and $-4$.
The product of these values is $8 \times (-4) = -32$.
Final Answer: -32

**### Question**
{Demonstration Question}
**### Thoughts and Answer**
{Demonstration Thoughts and Answer}
...

**### Question**
{Target Question}
**### Thoughts and Answer**
...
(Expecting "Final Answer" in the last line)

Figure 10: The instruction and one of the demonstration examples we use when prompting ChatGPT in the CoT setting.

---

**### Instruction**
You are given a math question.
You should generate a piece of python code to solve the problem.
Please show your thoughts in python codes.

**### Question**
Point $P$ lies on the line $x= -3$ and is 10 units from the point $(5,2)$. Find the product of all possible $y$-coordinates that satisfy the given conditions.
**### Solution Code**

```python
import math
# initilize the coordinates
x1 = -3
x2 = 5
y = 2
# initialize the distance
d = 10
# calculate 2 y-coordinates using the Pythagorean Theorem
y_possible1 = y + math.sqrt(d**2 - (x2 - x1)**2)
y_possible2 = y - math.sqrt(d**2 - (x2 - x1)**2)
# calculate the product
product = y_possible1 * y_possible2
print("The product of all possible y-coordinates is", product)
```

**### Question**
{Demonstration Question}
**### Solution Code**
{Demonstration Code}
...

**### Question**
{Target Question}
**### Thoughts and Answer**
...

Figure 11: The instruction and one of the demonstration examples we use when prompting ChatGPT in the PoT setting.

### Instruction
You are given a math question.
You could call WolframAlpha API to help you solve the question.
After seeing a question, you should first generate thoughts and think about how to call the API.
Generate "WOLFRAM:" in the last line of your response with appropriate inputs you'd like to inquiry.

### Question
Point $P$ lies on the line $x= -3$ and is 10 units from the point $(5,2)$.
Find the product of all possible $y$-coordinates that satisfy the given conditions.
### Response
Let the y-coordinate of point $P$ be denoted by $y$.
Since $P$ lies on the line $x=-3$, we know that the coordinates of $P$ are $(-3,y)$.
Since $P$ is 10 units from the point $(5,2)$, we can use the distance formula to write an equation:
$\sqrt{(-3-5)^2+(y-2)^2}=10$
We can call wolframalpha to solve this equation
WOLFRAM: solve sqrt((-3-5)^2+(y-2)^2)=10

### Question
{Demonstration Question}
### Response
{Demonstration Response}
...

### Question
{Target Question}
### Response
...
(Expecting "WOLFRAM" in the last line)

Figure 12: The instruction and one of the demonstration examples we use when prompting ChatGPT in the Tool Use setting. This figure shows the first part about WolframAlpha inquiry.

### Instruction
You are given a math question.
You have just called WalframAlpha API to help you solve the question.
Please continue to generate your final numerical answer with the return from WalframAlpha API as reference.
If There is an error return from the API, you could continue you thought step by step and give your final answer.
Generate "Final Answer:" in the last line of with your final numerical answer.

### Question
Point $P$ lies on the line $x= -3$ and is 10 units from the point $(5,2)$.
Find the product of all possible $y$-coordinates that satisfy the given conditions.
### Response
Let the y-coordinate of point $P$ be denoted by $y$.
Since $P$ lies on the line $x=-3$, we know that the coordinates of $P$ are $(-3,y)$.
Since $P$ is 10 units from the point $(5,2)$, we can use the distance formula to write an equation:
$\sqrt{(-3-5)^2+(y-2)^2}=10$
We can call wolframalpha to solve this equation
WOLFRAM: solve sqrt((-3-5)^2+(y-2)^2)=10
### WolframAlpha Return
[{'@title': 'Input interpretation', 'subpod': {'@title': '', 'plaintext': 'solve sqrt((-3 - 5)^2 + (y - 2)^2) = 10'}}, {'@title': 'Results', '@primary': 'true', 'subpod': [{'@title': '', 'plaintext': 'y = -4'}, {'@title': '', 'plaintext': 'y = 8'}], ...]
### Response Continue
From the WolframAlpha return, we see that there are two possible values for $y$, namely $y=-4$ and $y=8$. Therefore, the product of all possible $y$-coordinates is $-4 \cdot 8 = -32$.
Final Answer: -32

### Question
{Demonstration Question}
### Response
{Demonstration Response Same with the First Part}
### WolframAlpha Return
{Demonstration Return Dictionary from WolframAlpha API}
### Response Continue
{Demonstration of Final Answer Extraction}
...

### Question
{Target Question}
### Response
{LLM Response exactly in the First Part}
### WolframAlpha Return
{WolframAlpha Return from the First Part}
### Response Continue
...
(Expecting "Final Answer" in the last line)

Figure 13: The instruction and one of the demonstration examples we use when prompting ChatGPT in the Tool Use setting. This figure shows the second part about answer retrieving.

Figure 14: The instruction and one of the demonstration examples we use when prompting ChatGPT in the CRE-ATOR setting. This figure shows the prompts applied in the Creation stage.

### Instruction
You are given a math question, and the related tools that may help you solve the question.
Now you need to write python code to call the tool to solve the problem.
Read carefully about the tool's parameters, returns and how to use the tool.
Please print out your answer and any information you think it's important.

```python
# Given point, distance and center
x = -3
distance = 10
center = (5,2)
# Call the function which will return a list of y coordinates
ys = possible_ys(x=x, dist=distance, center=center)

# Calculate product of all possible y-coordinates
product = 1
# Check for the other possible y-coordinate
for y in ys:
    product *= y

# Print the answer
print("The product of all possible y-coordinates is", product)
```

### Question
{Demonstration Question}
### Tool
{Demonstration of Tools Created in Creation Stage}
### Solution
{Demonstration Solution Code to Print the Answer}
...

### Question
{Target Question}
### Tool
{The Tools Generated in Creation Stage}
### Solution
...
(Expecting Python Code Block)

Figure 15: The instruction and one of the demonstration examples we use when prompting ChatGPT in the CRE-ATOR setting. This figure shows the prompts applied in the Decision stage.

### Instruction
Your original codes have met some errors when executing, please rectify your codes.
You should generate the whole new piece of code (including the tools and the decision call), instead of generating a snippet.
Wrap your code in ```` ```python\n ... \n``` ```` to make it a one whole code block.
Pay attention to the conditions in question before modifying.
Your goal is answer the question correctly instead of just execute the code successfully.
If necessary, you can also generate a whole new tool, changing your way of thinking to solve the problem.

### Question
Please calculate the square root of a and b to get c, where a = 4 and b = 9.
### Original

```python
def calculate_c(a, b):
    """
    The function calculate c, which is the square root of a
    multiply b
    """
    c = math.sqrt(a * b)
    return c
a = 4
b = 9
c = calculate_c(a, b)
print("The value of c is", c)
```

### Error
Traceback (most recent call last):
  File "test_code.py", line 9, in <module>
    c = calculate_c(a, b)
  File "test_code0.py", line 5, in calculate_c
    c = math.sqrt(a * b)
NameError: name 'math' is not defined
### Rectification
The math is not defined, so I should add the math module before the tool.
In detail, I should add `import math` at the very beginning.

```python
# Modify here to import at the beginning
import math
def calculate_c(a, b):
    """
    The function calculate c, which is the aquare root of a
    multiply b
    """
    c = math.sqrt(a * b)
a = 4
b = 9
c = calculate_c(a, b)
print("The value of c is", c)
```

### Question
{Demonstration Question}
### Original
{Demonstration of Tools and Decisions in Previous Stages}
### Error
{Demonstration Error Traceback from Execution Stage}
### Rectification
{Demonstration Rectified Tools / Decision with Thoughts}
...

### Question
{Target Question}
### Original
{Tools and Decisions Made by LLM in Previous Stages}
### Error
{Traceback from Execution Stage}
### Rectification
...

Figure 16: The instruction and one of the demonstration examples we use when prompting ChatGPT in the CRE-ATOR setting. This figure shows the prompts applied in the Rectification stage.