# OpenReview forum: "CREATOR: Tool Creation for Disentangling Abstract and Concrete Reasoning of Large Language Models"
_EMNLP/2023/Conference — EMNLP 2023 Findings_

### Official Review · Reviewer_DLxW · 2023-07-28

**Soundness:** 3

**Excitement:**

3: Ambivalent: It has merits (e.g., it reports state-of-the-art results, the idea is nice), but there are key weaknesses (e.g., it describes incremental work), and it can significantly benefit from another round of revision. However, I won't object to accepting it if my co-reviewers champion it.

**Missing References:**

LLMs for tool-use have been explored in other areas. I don’t think it’s a requirement to include them, but if there is space to here is an example:

Wang et al. - Voyager: An Open-Ended Embodied Agent with Large Language Models - https://arxiv.org/abs/2305.16291

**Paper Topic And Main Contributions:**

This work presents Creator, a framework which utilizes LLMs (ChatGPT in this case) to solve mathematical and tabular word problems. The authors discretize the framework into 4 separate stages: (1) creation of a programmatic tool, (2) deciding how to use the tool for the current problem, (3) executing a program with the chosen tools, (4) rectifying any syntactic errors in the program, or if none then returning the output of the program as the answer.

The authors evaluate their framework on 2 existing benchmarks (MATH and TabMWP), and compare with existing methods (ChatGPT w/ and w/out chain of thoughts, program of thoughts, ChatGPT + WolframAlpha) finding that their method improves performance above the baselines.

**Questions For The Authors:**

Question A: How does the framework determine the number of tools to create for each query?

Question B: Is tool re-use available in your framework? What I’ve understood from your figures is that a new “tool” is generated for every query, and never gets reused. However, (2) on lines 70-74 suggests that your framework should be allowing the system to “benefit from finding robust commonalities among multiple similar questions”.

Question C: Section 5 seems to discuss an experiment about tool re-use, but the settings are very unclear. How did the model determine which tools to re-use? Given a query, how does the framework allow the model to access the previously created tools? I wasn't able to find any details about this in section 5.

Question D: Figure 2 shows that rectification will provide feedback for the tool creation, but section 3.4 makes it sound like the rectification will only be used to , but isn’t it possible that errors are thrown because of code from the program generated in the execution step. Is that case addressed in your framework?

Question E: In Table 3 you show an execution-accuracy gap of between 5%-9%, and I think some insights into the failure cases would be very insightful here. Are the tools not working as intended (semantic errors)? Is the model incorrectly using the tools? Is the model incorrectly translating the problem into the program in the execution step?

**Reasons To Accept:**

This paper presents a framework to solve a problem that (in my view) is currently a hot topic in the community. Tool use is a relatively new topic in NLP, and including this paper in the conference will be a benefit to those within the NLP community. Additionally, I believe that methods similar to this may be of interest to other communities as well (eg. neuro-symbolic AI, RL).

**Reasons To Reject:**

The authors introduce a new dataset (Creation Challenge) which is underspecified within the paper (1-2 paragraphs in the main body and 1 additional paragraph in the appendix) and therefore difficult to evaluate the validity of some of the findings. It’s possible that I missed something, but I wasn’t able to find any statistics about the created dataset (diversity of tools, types of questions, etc.)

Shortcoming #2 Fragility in reasoning (lines 70-74) is a challenge that the authors claim to address in this work (lines 79-80), which leads a reader to expect that their framework will not do "reasoning on the fly case-by-case". There seems to be a small scale experiment where tool re-use is allowed, but the experimental setting is very underspecified (for example, how does the model determine which tool to use?) The main results of this work don't seem to include tool re-use (to the best of my understanding), leading to a gap between the paper's claims and what actually exists in this work. If the main results do not include tool re-use then I believe that this work is misrepresenting itself by it's claim of reusing tools, and should be altered before it can be published in this venue.

**Reproducibility:**

2: Would be hard pressed to reproduce the results. The contribution depends on data that are simply not available outside the author's institution or consortium; not enough details are provided.

**Reviewer Confidence:**

4: Quite sure. I tried to check the important points carefully. It's unlikely, though conceivable, that I missed something that should affect my ratings.

---

> ### Author Rebuttal · Authors · 2023-08-28
>
> Thank you very much for your reviews. Large language models and their cooperation with tools is indeed a new topic of interest, and we take a step further from the usage of tools to the creation of tools, which emphasizes the creativity of current large language models, and their creativity’s potential in solving complex problems. Here are the responses to some of your concerns.
>
> > **Q1:** I wasn’t able to find any statistics about the created dataset
>
> Please refer to our response to Reviewer #1 about the necessity, novelty, and correctness of both the Creation Challenge as a benchmark and the Tool Transfer dataset applied in Sec. 5.1. Here is a more detailed statistic about the Creation Challenge dataset.
>
> |       Types       | Ratio |                       Examples                        |
> | :---------------: | :---: | :---------------------------------------------------: |
> |   Math-Related    |  30%  | geometric calculation, math application problem, etc. |
> | Finance and Money |  28%  |  interest calculator, investment, earning rate, etc.  |
> |      Physics      |  18%  |  problems of fluid, voltage, energy efficiency, etc.  |
> |  Data Processing  |  6%   |    search for value, statistic of variables, etc.     |
> |      Others       |  18%  |       population growth, BMI calculation, etc.        |
>
>
> Our data consist of problems of diverse types and domains, and include the problems that necessitate tool creation to solve under real-world scenarios. We will also include these statistics in a later updated version of this paper, and modify the structure to make our introduction of the dataset clearer.
>
> > **Q2:** The main results of this work don't seem to include tool re-use, leading to a gap between the paper's claims and what actually exists in this work.
>
> We would like to refer to the experiment in Sec. 5.1 for the testing of reusability of tools created under the CREATOR framework. We have separated the dataset we apply for Table 2-4 and Tool Transfer mainly due to:
>
> 1. The main experiments focus more on evaluating language models through distinct and novel questions in order to show tool creation’s strong capacity. Therefore, the tools are naturally less reusable, which is why we created a separate dataset for a more controlled and precise evaluation of transferability especially in Sec. 5.1. (Please also refer to the response to Reviewer #1’s Q4 for more details.)
>
> 2. The experiments on transferability contain more questions of the same type, as the three questions in the same set share the same core knowledge, which serves as a fairer test bed. In addition, the Tool Create Normal Acc we present in Table 5 is the result derived applying the CREATOR framework (normal indicates without the transfer of tool), and all the tools in this experiment are created under the CREATOR framework.
>
> The main experiment and the tool transfer experiment aim to show different aspects of advantages through tool creation, and they are separated for different purposes. We will clarify the separation of different settings at the beginning of Sec. 4 and 5, and further improve our writing in the introduction to avoid misunderstanding in the next version of the update.
>
> > **Q3:** Missing References: LLMs for tool use have been explored in other areas.
>
> The article [1] represents a category of studies focusing on AI agents and their acquisition of skills. The extension of the skill library under game scenarios introduced in [1] is analogous to the construction of the tool library for retrieval, the core idea of which is more similar to [2] in tool learning. However, our study focuses more on the model’s creativity itself instead of the acquisition of skills from the environment. Thanks for the suggestion. We will add the reference and state the differences in the later updated version of the paper.
>
> > **Question A:** How does the framework determine the number of tools to create for each query?
>
> Our framework does not explicitly determine the number of tools that should be generated. We picture this should be determined by the model after being given the query. Besides, the model should also automatically decide how to apply multiple tools and in what form. In Figure 9 in the Appendix, we showcase the different levels of tool creation, all of which actually involve multiple tools that the model created. For instance, if the model decides to use nested tool calls in its creation, then at least 2 tools must be created in order to fulfill the query’s requirement.
>
>
> > **Question B:** Is tool re-use available in your framework?
>
> The tools created under the CREATOR framework could be reused, as we have shown in the experiment in Sec. 5.1. The reason for the separation of experiments is stated in the response to Q2.
>
> The primary purpose of tool transfer is to avoid repetition, and the created tools are particularly helpful when faced with queries with similar core knowledge, such as the solving of quadratic functions, etc. This motivates us to design the scenarios in Sec. 5.1, where we test if the creation of one tool can simply apply to other similar queries.
>
> Since the process of "creation" consumes a significant amount of computational resources, reusing tools can reduce computational costs. Its importance lies in the fact that when there are numerous repetitive tasks, or tasks that share the same core knowledge, repeatedly reasoning on the fly is not as efficient as designing a tool and then reusing it. This is also one of the motivations behind tool creation. We will further improve how we present the reuse of created tools and the underlying logic among different experimental settings in the next version of the paper.
>
> > **Question C:** How did the model determine which tools to re-use? Given a query, how does the framework allow the model to access the previously created tools?
>
> In the experiment, we introduce in Sec. 5.1 (not the pipeline we illustrate in Figure 2, which is for the main experiments), the model is first given only one question and asked to create related tools that could be applied to solve this one question. After the creation, we present the model to the other two scenarios in the same set that share the same core knowledge as the first one. The model is then asked to apply its previously created tools (not specifically tailored for the current two new scenarios) to solve the current two new problems, choosing which tools to reuse and deciding on how to use them all by itself. The transferability is measured by the success of solving both new problems by applying the previously created tools. Our results in Table 5 show how using the transferred tool could even raise the performance than creating a new one.
>
>
> > **Question D:** Isn’t it possible that errors are thrown because of code from the program generated in the execution step. Is that case addressed in your framework?
>
> Thanks for bringing our attention to this issue. It is a possible situation and we have taken this situation into consideration. If the tool doesn't raise an execution error, and the only problem lies on the decision call, then in the next round of rectification, the created tool given to the model will remain the same (skip the creation stage in the next iteration), and only modification on decision call will be performed. We will add a more detailed explanation about it in the first paragraph of Sec. 3.4 to avoid the confusion that might be caused.
>
> > **Question E:** In Table 3 you show an execution-accuracy gap of between 5%-9%, and I think some insights into the failure cases would be very insightful here.
>
> Firstly, the error will not be caused by the compilation error of created tools as the validity of the tools is rigorously checked before being given to the model for application.
>
> After investigating the errors of wrong execution outputs, we discover the errors can be divided into the following categories, and we present the new case studies as follows:
>
> 1.**The erroneous interpretation of the problem:** In this category, the information provided in the problem is wrongly interpreted, resulting in the wrong initial condition during the execution stage.
>
> >**Question:**  Principal Navarro reported the state test scores from some students at her school. How many students scored at least 45 points but fewer than 59 points?
> >
> >**Table:**
> >```
> Stem | Leaf
> 2 | 0
> 3 | 4
> 4 | 0, 3, 6
> 5 | 1, 1, 5
> 6 | 0, 4
> >```
> >**Created Tool:**
> >……
> >
> >**Decision:**
> >```python
> > # Initialize the table
> df = pd.DataFrame({
>     'Stem': [2, 3, 4, 4, 4, 5, 5, 5, 6, 6],
>     'Leaf': [[0], [4], [0, 3, 6], [0, 3, 6], [0, 3, 6], [1, 1, 5], [1, 1, 5], [1, 1, 5], [0, 4], [0, 4]]
> })
> ……
> >```
> >**Analysis:** In this case, the table given in the question is wrongly interpreted. Though the created tool functions as intended, the wrong initial condition provided as the direct interpretation of the table leads to a wrong final answer.
>
> 2.**Misalignment between tool’s purpose and decision call:** In this category, the model uses the tool in a way not intended as the tool is created, thus causing chaos in the logic of problem-solving.
>
> > **Question:** Each shelf has 10 hats. How many hats are on 5 shelves?
> >
> >**Table:**
> >```
> Number of shelves | Number of hats
> 1 | 10
> 2 | 20
> 3 | 30
> 4 | 40
> 5 | ?
> >```
> >**Created Tool:**
> >```python
> def calculate_hats_on_shelves(df, num_shelves):
>     """
>     This function takes in the number of shelves to calculate the number of hats.
>     """
> ……
> >```
> >**Decision:**
> >```python
> ……
> ># Calculate the number of hats on 5 shelves
> num_hats = calculate_hats_on_shelves(df, 5)
> ># Calculate the total number of hats
> total_hats = num_hats * 5
> ……
> >```
> >**Analysis:** In this case, the tool created is used to calculate the number of hats, which is valid for solving the problem. However, the model instead wrongly thinks the output of the tool is the number of hats on one shelf (which is 10), and further multiplies five to the tool’s output, which increases unnecessary complexity and leads to wrong answers.
>
> From the case studies, we discover problems like misinterpretation and misalignment are actually causing more errors than a wrong tool created. This sheds light on a promising future research direction to bridge the gap. Thanks for bringing our attention to this error analysis, and we will include this as an experimental case study in the later updated version of the paper.
>
> >[1] Wang G, Xie Y, Jiang Y, et al. Voyager: An open-ended embodied agent with large language models[J]. arXiv preprint arXiv:2305.16291, 2023.
> >
> >[2] Paranjape B, Lundberg S, Singh S, et al. ART: Automatic multi-step reasoning and tool-use for large language models[J]. arXiv preprint arXiv:2303.09014, 2023.

---

### Official Review · Reviewer_PJDS · 2023-08-05

**Soundness:** 3

**Excitement:**

3: Ambivalent: It has merits (e.g., it reports state-of-the-art results, the idea is nice), but there are key weaknesses (e.g., it describes incremental work), and it can significantly benefit from another round of revision. However, I won't object to accepting it if my co-reviewers champion it.

**Paper Topic And Main Contributions:**

This paper is about the CREATOR framework, which enables Large Language Models (LLMs) to create their own tools and improve their performance on complex problem-solving tasks. The paper addresses the limitations of current approaches in utilizing a limited number of tools and the fragility of reasoning on the fly case-by-case. The main contributions of the paper include the development and evaluation of the CREATOR framework on challenging benchmarks, the Creation Challenge dataset, and the demonstration of how leveraging LLMs as tool creators facilitates knowledge transfer and drives us closer to the next frontier of artificial intelligence. The paper falls under the category of NLP engineering experiment and approaches for data- and compute efficiency.

**Reasons To Accept:**

The strengths of this paper include its novel approach to empowering Large Language Models to create their own tools, its evaluation on challenging benchmarks, and its demonstration of the benefits of leveraging LLMs as tool creators for knowledge transfer and improved performance. The paper also presents the Creation Challenge dataset, which emphasizes the necessity and benefits of LLMs' tool creation ability. If this paper were to be presented at the conference or accepted into Findings, the main benefits to the NLP community would be the introduction of a new framework for improving LLMs' performance on complex problem-solving tasks, the availability of a new dataset for evaluating LLMs' tool creation ability, and the potential for driving us closer to the next frontier of artificial intelligence. The paper's contributions to NLP engineering experiment and approaches for data- and compute efficiency would also be valuable to the community.

**Reasons To Reject:**

The paper acknowledges that the execution ability and correctness of the LLM's tool creation potential lack proper evaluations and remain questionable. Therefore, the evaluation of the LLM's capabilities may not be comprehensive enough to support the claims made in the paper.

**Reproducibility:**

3: Could reproduce the results with some difficulty. The settings of parameters are underspecified or subjectively determined; the training/evaluation data are not widely available.

**Reviewer Confidence:**

4: Quite sure. I tried to check the important points carefully. It's unlikely, though conceivable, that I missed something that should affect my ratings.

---

> ### Author Rebuttal · Authors · 2023-08-28
>
> Thank you very much for your review. We also believe our work will bring benefits to the whole NLP community by bringing out a new framework for complex task-solving through tool-creation, and a new evaluation standard on the large language model’s creativity. Here is the may address your concerns:
>
> > **Q1:** The execution ability and correctness of the LLM's tool creation potential lack proper evaluations and remain questionable. Therefore, the evaluation of the LLM's capabilities may not be comprehensive enough to support the claims made in the paper.
>
> First of all, thank you for raising this interesting concern. As we state in the limitation section, in this work we focus on tackling tasks that current evaluation technologies allow us to. For tasks where automatic evaluations are still immature or undeveloped, such as the evaluation of a complex pipeline (e.g. the construction of a webpage, the analysis based on a database, etc.), conducting such evaluations requires developing a whole set of new techniques. This is still an open question and is beyond the scope of this work. Therefore our work focuses on tasks such as math reasoning and tabular reasoning, which follow common practices of prior works [1] [2] [3] and allow for efficient and controlled evaluation.
>
> >[1] Schick T, Dwivedi-Yu J, Dessì R, et al. Toolformer: Language models can teach themselves to use tools[J]. arXiv preprint arXiv:2302.04761, 2023.
> >
> >[2] Paranjape B, Lundberg S, Singh S, et al. ART: Automatic multi-step reasoning and tool-use for large language models[J]. arXiv preprint arXiv:2303.09014, 2023.
> >
> >[3] Qin Y, Hu S, Lin Y, et al. Tool learning with foundation models[J]. arXiv preprint arXiv:2304.08354, 2023.

---

### Official Review · Reviewer_yYH8 · 2023-08-15

**Soundness:** 3

**Excitement:**

3: Ambivalent: It has merits (e.g., it reports state-of-the-art results, the idea is nice), but there are key weaknesses (e.g., it describes incremental work), and it can significantly benefit from another round of revision. However, I won't object to accepting it if my co-reviewers champion it.

**Paper Topic And Main Contributions:**

This paper proposes a new prompting framework for creating and utilizing functions to solve math reasoning problems with Python programs. It separates abstract and concrete reasoning by designing Python functions to help solve the corresponding problem, then writing code to solve it by execution. Additionally, this paper introduces a dataset for evaluating the ability of LLMs to create tools. Experiments in solving challenging math problems show that the proposed method is effective. Additional analyses show that the created tools can be transferred across different scenarios.

**Questions For The Authors:**

A. The introduction to the newly created datasets `Creation Challenge` and `Tool Transfer` is somewhat brief and vague. Besides, as a benchmark, how to ensure the correctness of problems and references generated by LLMs?

B. The process of answer extraction is not well-presented, which may affect the reported scores for the MATH dataset. After inspecting the implementation, I think it may not be appropriate to try to match every float number in the printed information. Will it cause false recalls?

**Reasons To Accept:**

* The proposed method is well-motivated and intuitive. The idea of tool creation is novel compared to existing work.
* The proposed method achieves fairly good results on very challenging math problems, i.e., a subset of the MATH dataset.
* The paper is well-written and easy to follow.

**Reasons To Reject:**

* Utilizing a Python interpreter to solve math word problems (e.g., GSM8k, TabMWP) does not raise this concern, but in the MATH dataset, there are problems that are very easy to be solved with a Python program, yet need carefully designed construction for human without external tools. To give a simplified example that does not exist in the MATH dataset, to calculate the sum of 1 to 100, it is fairly easy to list a program that directly sum all the numbers, however, human will adopt the `sum of equivariant series` by understanding the property of the series. In fact, this paper discusses the different impacts of CoT for helping PoT-like methods in MWP and the MATH dataset, which may be evidence for this concern. I think PoT-like methods may not perform real math reasoning, but leverage a shortcut in solving non-MWP math problems. I think it deserves a discussion since this is one of the first works to adopt PoT-like methods in the MATH dataset.
* The claim that the created tools can be transferred across different questions is questionable. For example, in the prompt provided in the supplemental materials, some created tools containing specific conditions from the problem are obviously not transferable to other problems, such as `solve_equation` for the problem `If $3p+4q=8$ and $4p+3q=13$, what is $q$ equal to?`. Besides, there lack of a mechanism to verify whether a tool is transferable, or extract transferable tools from extensive solutions.
* The test set of the MATH dataset is not complete as explained in Appendix A.2, which also addresses a possible limitation of the proposed methods. I think it is not that hard to evaluate the full test set, as shown in the [implementation](https://github.com/openai/prm800k/blob/main/prm800k/grading/grader.py).
* The novelty and necessity of the new dataset are not well demonstrated.
* As a minor issue, the implementation of `Tool Use` may not be fair enough, since it only performs one time of calculation using the Wolfram API for each exemplar, as shown in the prompt.

**Reproducibility:**

5: Could easily reproduce the results.

**Reviewer Confidence:**

4: Quite sure. I tried to check the important points carefully. It's unlikely, though conceivable, that I missed something that should affect my ratings.

---

> ### Author Rebuttal · Authors · 2023-08-28
>
> Thank you very much for your reviews. LLM x Tool is indeed a relatively new topic in NLP and needs more future research. Previous research mostly focuses on tool-using, but we are among the first ones to propose using tool-creation in complex task-solving frameworks and as a measurement of LLM’s creativity. Here are the responses to some of your concerns.
>
> > **Q1:** This paper discusses the different impacts of CoT for helping PoT-like methods in MWP and the MATH dataset, which may be evidence for this concern. I think PoT-like methods may not perform real math reasoning, but leverage a shortcut in solving non-MWP math problems.
>
> Thank you for raising this insightful concern. We would like to refer you to Sec. 4.4 where we have discussed the difference between CoT and PoT logic. Though the solving of MATH problem may take a short-cut (e.g. the application of brute-force algorithms), it doesn’t hamper the validity of our framework, because:
> i) straightforwardness and preciseness are the attributes of code as the underlying medium that tool creation is taking advantage of;
> ii) large language models have their own way of conducting concrete reasoning through codes [1] in MATH, which may not necessarily be the same as they do in natural language;
> Thanks for raising this concern, and we will add these discussions in the next version of our update.
>
> > **Q2:** There lack of a mechanism to verify whether a tool is transferable, or extract transferable tools from extensive solutions
>
> Transferability is a unique novelty of tool creation that differentiates it from PoT or CoT methods. We have provided preliminary evidence from Table 5 that the created tools could be successfully transferred, and the resulting improvement of 15.3% proves it makes the reasoning more robust than simply conducting tool creation to solve the problems.
>
> In our experiment presented in Sec. 5.1, the choice of which transferred tool to use is decided by the model instead of the human. We also picture that the large language model should possess the ability to discern from mass amounts of tools, as the open-source model now possesses the ability to choose what tools to use from 16,000+ APIs [2]. Therefore, deciding which tools to use from a mass pool is more important than deciding the tool’s transferability itself, because any tool has the potential to be transferred as long as the scenario of queries are similar. Our CREATOR framework contributes to creating tools that possess transferability.
>
> The tool creation stage in the CREATOR framework necessitates that tools be created and applied in the solution, so these created tools can be directly used as potential sources of transfer, and no additional extraction is needed. We will put more discussion on these issues in the later updated version of this paper.
>
> > **Q3:** The test set of the MATH dataset is not complete as explained in Appendix A.2
>
> Thank you for this constructive comment. We haven’t used the entire test set because some answer formats are challenging for automatic evaluation (e.g. the matrix as answer), as explained in Appendix A.2. Following your provided code scripts, we conduct a more comprehensive analysis on all the data points from the MATH test set. Here are the updated results.
>
> |           Method            | Algebra (1187) | Counting & Probability (474) | Geometry (479) | Itmd. Algebra (903) | Number Theory (540) | Pre-Algebra (871) | Pre-Calculus (546) | Average (weighted) |
> | :-------------------------: | :------------: | :--------------------------: | :------------: | :-----------------: | :-----------------: | :---------------: | :----------------: | :----------------: |
> |      Vanilla (w/ CoT)       |     46.25      |            24.47             |     17.33      |        11.30        |        27.96        |       48.22       |        6.96        |       29.18        |
> |   PoT (w/o CoT, w/ Rec.)    |     55.86      |            51.05             |     32.15      |        27.91        |        56.11        |       66.48       |       26.19        |       46.72        |
> |   PoT (w/o CoT, w/o Rec.)   |     48.44      |            45.36             |     31.73      |        24.36        |        50.37        |       60.73       |       22.89        |       41.76        |
> |      Tool Use (w/ CoT)      |     43.13      |            26.16             |     19.42      |        11.85        |        34.81        |       48.45       |        9.52        |       27.62        |
> | CREATOR-Entangled (w/o CoT) |     56.95      |            53.80             |     30.90      |        32.56        |        56.85        |       65.90       |       23.26        |       47.62        |
> |      CREATOR (w/o CoT)      |     62.85      |          **59.28**           |   **37.58**    |      **35.88**      |      **63.70**      |     **70.26**     |     **27.47**      |     **52.74**      |
>
> For each method, we choose the setting (w/ or w/o CoT) in Table 2 with the highest accuracy for comparison. From the results, we could conclude CREATOR still achieves the highest accuracy on the weighted average and even on all 7 subtasks, which demonstrates the advantage of tool creation and disentanglement of reasoning. It should also be noted that the average for every method is lowered, potentially caused by the same challenging format matching problem (e.g. difficulty in matching the matrices as final answer). We will update all the results above and provide results on all settings in the later version of this paper.
>
> > **Q4:** The novelty and necessity of the new dataset are not well demonstrated. how to ensure the correctness of problems
>
> We agree on the importance of more clarification on these factors. We would like to refer the reviewer to lines 380 to 388, lines 487 to 492, and appendix B.1 and B.2 for a brief description of how we construct the data point and ensure novelty and diversity during task design. In the following, we elaborate with more details. We will include them in our final revision, and we will continue to improve our clarity in writing.
>
> **Necessity of Creation Challenge:** The motivation behind designing this dataset is that the existing benchmarks can mostly be solved by importing existing tools, and does not require creating novel tools to achieve high scores. Therefore, we have to design a new dataset with creativity being considered an important metric. We compare a question from the existing MATH dataset and our dataset for comparison.
>
> >**MATH Example 1:** What is $62_7+34_5$ when expressed in base 6?
> >
> >**Solution:** ```result_base6 = int(str(int('62', 7) + int('34', 5)), 6)```
> >
> >**Analysis:** The function of int() provided could solve the seemingly complex problem in one line.
>
> >**MATH Example 2:** If $\mathbf{A}^{-1} = \begin{pmatrix} 4 & 0 \\\\ 1 & -1 \end{pmatrix}$ and $\mathbf{B}^{-1} = \begin{pmatrix} 2 & -1 \\\\ -1 & 3 \end{pmatrix},$ then find the inverse of $\mathbf{A} \mathbf{B}.$
> >
> >**Solution:** ```inverse_of_AB = np.linalg.inv(A_inv @ B_inv)```
> >
> >**Analysis:** The numpy package as an existing tool could also easily solve the problem.
>
> >**Creation Challenge Example:** Suppose Alice deposited \\$1000 in a savings account that has an annual interest rate of 3%. Bob deposited \\$2000 in a savings account that has an annual interest rate of 5%. They both kept the money in their respective savings account for 10 years. How much more interest did Bob earn, compared to Alice?
> >
> >**Solution:** ```def calc_interest(initial_deposit: float, interest_rate: float, years: int) -> float: …```
> >
> >**Analysis:** The question necessitates the creation of tools for the convenience of reusing, and no existing tool or code package is directly relevant to the problem setting under real-world scenarios.
>
> **Novelty of Creation Challenge:** The Creation Challenge dataset we curated have several novelties (as a more detailed illustration to Appendix B.2, line 840-850):
> 1. Creation of new tools is required. The problems in the Challenge cannot be solved using existing tools or code packages directly. The problems are mostly based on real-world scenarios and necessitate a model's careful interpretation and creativity to reach the answer.
> 2. Diverse tool creation scenarios. We include various scenarios that necessitate the creation of tools in diverse domains, ranging from math, physics, finance, to problems in everyday life.
> 3. Different levels are included. For each data point, we include various meta-information about what tools are required, how the input and output of tools might be, etc. These contribute to varying levels of difficulty for the tool creation challenge, which makes the evaluation on creativity more flexible and fine-grained.
>
> **Necessity of Tool Transfer Data:** The data points in current datasets show great variance among individual questions, and the queries are not clustered based on the specific target knowledge needed, thus making the tests of tool transfer inappropriate. In order to achieve greater control and precision in measuring transferability, we aim to create a small-scale dataset for investigation
>
> **Novelty of Tool Transfer Data:** The tool transfer data we curated possess the following novelties (as a more detailed illustration to Sec. 4.2, line 380-388):
> 1. Shared core knowledge: The problems are clustered into a set of three, which share the same core knowledge. This provides a fair test bed for tool transfer, where the interference of the tool’s transferability itself or the ability of tool retrieval can be excluded.
> 2. Diverse real-world scenarios: Though the core knowledge is shared, the descriptions of problems in the same set possess diverse scenarios, which makes the transfer of the created tools also challenging to apply.
>
> **Correctness of Both:** Though the creation of problems is automatic with the help of LLMs, we also conduct vigorous supervision from two steps:
> 1. Self-consistency: For each problem generated, we feed it back for the model to generate the solution in different styles of reasoning as verifications, including reasoning through PoT and reasoning through CoT. Only the queries and answers that are consistently agreed upon by all verifiers will be retained.
> 2. Human supervision: For each retained problem, we also conduct human supervision to ensure the validity of potentially useful tools and the correctness in reasoning about their usage, thus ensuring the correctness of the final answer.
>
> > **Q5:** As a minor issue, the implementation of Tool Use may not be fair enough, since it only performs one time of calculation using the Wolfram API for each exemplar.
>
> We justify the Tool Use scenarios from the following perspectives:
> 1. After investigating the results in our earlier trials (in which we performed tests on randomly chosen queries from MATH), we found the usage of Wolfram API once is enough to solve roughly 85% of the problems as long as the query to API is designed properly.
> 2. Wolfram API has already implemented a complex heuristic algorithm for trying to solve a problem over multiple reasoning steps. It also provides a step-by-step solution [3] which further demonstrates its multi-step reasoning ability.
> Our comparison of Tool Use can also be viewed as an illustration that automatic tool creation has the potential to win over complex manual tool creation by experts.
>
> > **Q6:** The process of answer extraction is not well-presented
>
> In the latest updated version, we apply the evaluation method from the script, which turns the string into sympy expression format and conducts the comparison. This is especially crucial for the evaluation of the MATH dataset, as its queries possess various latex symbols in the answer which may cause confusion during the evaluation.
>
> For the other two datasets, the answers are all in float numbers, and we extract the answer exactly based on the fixed format of response (e.g. we only retrieve the model’s output after it outputs “Final Answer:”). Through this approach, we guarantee the answer we extract is not arbitrary numbers in the response, but what the model exactly believes in as the final answer.
>
> In addition, we also manually inspected the answers and verified that no numbers are incorrectly extracted. Though we notice this might not be a formal way, we guarantee there exists no false recall or random output, thus ensuring the reproducibility of our results. We will later update all our results using more formal comparison methods (e.g. the one we now apply for MATH), and state this in a clearer way in Sec. 4.1.
>
> >[1] Chen W, Ma X, Wang X, et al. Program of thoughts prompting: Disentangling computation from reasoning for numerical reasoning tasks[J]. arXiv preprint arXiv:2211.12588, 2022.
> >
> >[2] Qin Y, Liang S, Ye Y, et al. Toolllm: Facilitating large language models to master 16000+ real-world apis[J]. arXiv preprint arXiv:2307.16789, 2023.
> >
> >[3] Wolfram|Alpha Examples: Step-by-Step Solutions (wolframalpha.com)

---

### Meta-Review · Area_Chair_g2Wo · 2023-09-18

**Recommendation:** 3

**Metareview:**

This paper presents an innovative approach to enhance the capabilities of Large Language Models (LLMs) by introducing the CREATOR framework, which allows LLMs to create their own tools using documentation and code realization. The results on MATH and TabMWP benchmarks, as well as the introduction of the Creation Challenge dataset, highlight the potential of this approach to revolutionize problem-solving paradigms. The reviewers concur that the method is intriguing and valuable in the context of LLMs as agents. However, concerns have been raised about the experimental setting and its ability to support the claims made in the paper. In light of these concerns, we recommend accepting the paper to Findings.

---

### Decision · Program_Chairs · 2023-10-07

**Decision:**

Accept-Findings

**Comment:**

This paper presents an innovative approach to enhance the capabilities of Large Language Models (LLMs) by introducing the CREATOR framework, which allows LLMs to create their own tools using documentation and code realization. The results on MATH and TabMWP benchmarks, as well as the introduction of the Creation Challenge dataset, highlight the potential of this approach to revolutionize problem-solving paradigms. The reviewers concur that the method is intriguing and valuable in the context of LLMs as agents. However, concerns have been raised about the experimental setting and its ability to support the claims made in the paper. In light of these concerns, we recommend accepting the paper to Findings.